# Evaluation of the Hamburg-Glasgow Classification in Pancreatic Cancer: Preoperative Staging by Combining Disseminated Tumor Load and Systemic Inflammation

**DOI:** 10.3390/cancers13235942

**Published:** 2021-11-25

**Authors:** Thaer S. A. Abdalla, Valeria Almanfalouti, Katharina Effenberger, Faik G. Uzunoglu, Tarik Ghadban, Anna Dupreé, Jakob R. Izbicki, Klaus Pantel, Matthias Reeh

**Affiliations:** 1Department of General, Visceral and Thoracic Surgery, University Medical Center Hamburg-Eppendorf, 20251 Hamburg, Germany; thaer.abdalla@uksh.de (T.S.A.A.); v.almanfalouti@gmail.com (V.A.); f.uzunoglu@uke.de (F.G.U.); t.ghadban@uke.de (T.G.); a.dupree@uke.de (A.D.); izbicki@uke.de (J.R.I.); 2Department of Tumor Biology, University Cancer Center Hamburg, Center for Experimental Medicine, University Medical Center Hamburg-Eppendorf, 20251 Hamburg, Germany; keffenberger@uke.de (K.E.); pantel@uke.de (K.P.)

**Keywords:** liquid biopsy, pancreatic ductal adenocarcinoma, tumor staging, disseminated tumor cells

## Abstract

**Simple Summary:**

Despite the great achievements in pancreatic ductal adenocarcinoma (PDAC), identification of patients who will suffer rapid disease relapse and progression is not perfect, when definitive histology for tumor staging is not available. The Hamburg Glasgow Classification combines tumor cell dissemination in the bone marrow and systemic inflammatory response into a preoperative staging system. In this work, we assessed the Hamburg Glasgow Classification in potentially resectable PDAC as a prognostic classification for overall and progression-free survival and compared it to the UICC-TNM classification with promising results.

**Abstract:**

This study aims to compare the Hamburg Glasgow Classification (HGC) to Union for International Cancer Control (UICC) classification in patients with pancreatic ductal adenocarcinoma (PDAC). As adequate tumor classification is only possible after tumor resection and histological evaluation, only 20% of patients with PDAC receive accurate tumor staging. Thus, an accurate preoperative staging system is still missing but urgently needed. Systemic inflammation and tumor dissemination are important factors regarding the oncological outcome. HGC integrates both into a preoperative staging system, by combining C-reactive protein (CRP), albumin, and disseminated tumor cells (DTC) in the bone marrow. In this prospective study, 109 patients underwent surgical exploration for suspected PDAC. All patients underwent a preoperative bone marrow aspiration for DTC detection. HGC showed significant preoperative risk stratification for overall survival (OS) (*p*-value < 0.001) and progression-free survival (PFS) (*p*-value < 0.001). These results were comparable to the UICC survival stratification for OS and PFS (*p*-value = 0.001 and 0.006). Additionally, in non-metastatic PDAC, HGC III-IV was associated with shorter OS and PFS (*p*-value < 0.001, respectively) when compared to HGC I-II. Therefore, the HGC is a promising preoperative prognostic staging classification for accurate and simple outcome stratification in patients with PDAC.

## 1. Introduction

Pancreatic ductal adenocarcinoma (PDAC) is the 7th leading cause of cancer death in industrialized countries [1]. It is characterized by rapid progression, early metastasis, and low sensitivity for radiation and chemotherapy with a low 5-year survival rate of 10% [2]. Surgical resection remains the only curative modality, however, due to early local progression and metastasis, primary resection is only possible in 20% of the patients at presentation [3].

Accumulating evidence indicates that early disease progression in PDAC is due to the presence of micrometastasis or the presence of disseminated tumor cells (DTC) at distant sites at the time of diagnosis. The presence of DTC in bone marrow has proven to be a prognostic indicator for progression-free and overall survival in PDAC. This suggests that the bone marrow might be a reservoir for disseminating PDAC cells [4,5].

However, patients’ outcomes are not solely based on tumor-related factors such as tumor size, grade, lymph node involvement, and systemic tumor dissemination [4,5,6] but are also dependent on patient-related factors. One of the most important is the systemic inflammatory response (SIR); this reflects the immune system’s response against the proliferation and survival of tumor cells, which affects its ability to spread through tumor-induced angiogenesis, and metastasis [7,8,9].

Over the last few decades, the American Joint Committee on Cancer (AJCC) created the TNM staging system, which is the most commonly used staging system to differentiate between groups with different survival outcomes [10]. However, accurate stratification is only possible after tumor resection and histological analysis. Since only a minority of tumors are considered resectable at presentation, other risk-stratification staging systems are required for appropriate guidance of treatment.

The Glasgow Prognostic Score (GPS) is based on two acute phase proteins, albumin, and C-reactive protein (CRP). This represents an indicator of SIR, which has been proven to be a useful tool for risk stratification in patients with colorectal, esophageal, and lung cancer [11,12].

Merging the DTC status with the GPS combines SIR and the disseminated tumor load forming a preoperative staging classification, namely the Hamburg-Glasgow Classification (HGC). This can stratify patients into different risk groups before oncologic resection. Previously, we assessed HGC in non-metastatic esophageal cancer which resulted to be a strong, significant, and independent predictor of overall survival and disease-free survival [13].

This study aims to assess the Hamburg-Glasgow Classification in patients with PDAC as a prognostic classification for overall and progression-free survival.

## 2. Materials and Methods

### 2.1. Patient Characteristics

All patients enrolled in this study underwent surgical exploration for suspected PDAC at the University Hospital Hamburg-Eppendorf. The indication for treatment was according to the German S3 guidelines [14]. The study was approved by the Medical Ethical Committee, Hamburg, Germany (PV 3548). Informed consent was obtained from all patients before study inclusion.

Preoperative routine workup included patient’s history, physical examination, routine blood tests, thoracic and abdominal CT-scans, in selected cases endoscopy with fine-needle aspiration. In the case of non-resectable PDAC, the cTNM stage was used to classify patients in their respective UICC stages. The database included 194 patients who had bone marrow aspiration at the time of surgery between July 2004 and March 2010. We retrospectively included only patients with histological evidence of pancreatic ductal adenocarcinoma (PDAC). Therefore, 16 patients with distal cholangiocarcinoma and 22 patients with benign pancreatic lesions were excluded. After excluding 47 for missing preoperative Albumin or CRP, 109 patients were enrolled in this study. Post-operative follow-up was conducted according to the S3 German guidelines [14] at 3-month intervals for the first 2 years, including physical examination, abdominal ultrasonography, and computed tomography of the chest and abdomen. Information regarding postoperative chemotherapy was gathered retrospectively and was available only for 51 patients. Data regarding chemotherapeutic regimens, dosage, frequency of application, and response to chemotherapy were not available.

### 2.2. Disseminated Tumor Cells in the Bone Marrow

In this study, all patients who underwent surgical exploration for PDAC received a preoperative bone marrow aspiration. This was done under aseptic measures. 10 mL of bone marrow was taken from the upper right iliac crest. Mononuclear cells were enriched using the Ficoll gradient. Disseminated tumor cells were detected using an anti-cytokeratin antibody (A45-B/B3). This immunocytochemical assay for DTC detection in bone marrow has a false-positive rate of a maximum of 1% in control cases [15]. All stained slides were processed and screened by automated screening devices (ACIS™ system and Ariol™ system). All images were finally reviewed by two independent investigators for their microscopic morphological features. This protocol has been previously established and described in detail for DTC detection in PDAC [5].

### 2.3. Hamburg Glasgow Classification

The HGC was previously evaluated in patients with esophageal cancer by Reeh et al. [13]. The components of the Glasgow Prognostic Score (Albumin and CRP) were merged with the detection of DTC into HGC, which resulted in four prognostic groups (Table 1). Abnormal preoperative results were defined as follows: (i) positive DTC status (≥1 DTC) (ii) elevated CRP (>10 g L^−1^), and (iii) hypoalbuminemia (<35 g L^−1^) (Table 1).

### 2.4. Statistical Analysis

For statistical analysis, SPSS 26 for Windows (Armonk, NY, USA) was used. Descriptive statistics were used to describe patient baseline characteristics. The association between the HGC and clinicopathological parameters was evaluated using the X^2^ test. Survival curves for progression-free and overall survivals of the patients were plotted using the Kaplan–Meier method and analyzed using the log-rank test. Results are presented as median survival in months with a 95% confidence interval (CI) and the number of patients at risk.

The overall survival was computed as the period from the date of surgery to either the date of death or last follow-up, whichever occurred first. The progression-free survival was defined from the date of surgery to the date of evidence of recurrence, last follow-up, or date of death, whichever occurred first.

Multivariate analysis using Cox regression was used to assess the independent prognostic influence of HGC and other parameters on progression-free survival and overall survival. Results are presented as hazard ratio (HR) with 95% CI. Significant statements refer top-values of two-tailed tests that were *p* < 0.05.

## 3. Results

### 3.1. HGC and Patient Characteristics

One hundred and nine patients following surgical exploration for suspected resectable PDAC with available DTC, albumin, and CRP status were included. Tumor resection was carried out in 82 patients, this included 65 pancreatic head resections, 13 distal pancreatic resections, and 4 total pancreatectomies. Of the 109 patients, 57 (52.3%) were male. The median age was 65 (range 42–84 years). This cohort included 3 patients with UICC stage I, 49 patients with UICC stage II, 14 patients with UICC stage III, and 43 patients with UICC stage IV. DTCs were detected in 20 patients preoperatively. The overall frequency of DTC was 1–3 per 2 × 10^6^ mononuclear cells. We assessed the association of HGC with sex, age, and the following histopathological parameters: tumor grade, tumor size, lymph node stage, metastatic status, resection margin, and post-operative Union for International Cancer Control (UICC) TNM classification. There was no association between the above-mentioned parameters and the HGC (Table 2). Furthermore, no complications were reported after the bone marrow aspirations.

### 3.2. Univariate Analysis

The median survival time was 10.8 months. Hamburg-Glasgow classification showed significant preoperative risk stratification for overall and progression-free survival (*p*-value < 0.001, respectively). These results were comparable to the survival stratification of the UICC TNM classification for OS (*p*-value = 0.001) and PFS (*p*-value 0.006) (Figure 1 and Figure 2). In patients with non-metastatic PDAC, HGC III–IV was associated with shorter OS and PFS (*p*-value < 0.001 each) compared to HGC I–II (Figure 3). In metastatic PDAC (UICC IV), HGC III–IV had worse OS (*p*-value = 0.03) and PFS (*p*-value = 0.025) compared to HGC I–II (Appendix A).

The presence of distant metastases at the time of exploration was associated with shorter OS (*p*-value < 0.001) and shorter PFS (*p*-value = 0.002) (Appendix A). Patients without tumor resection had comparable OS (*p*-value = 0.190) and shorter PFS (*p*-value = 0.013) to R2 resections but significantly shorter OS (*p*-value < 0.001 and 0.001) and PFS (*p*-value = 0.004 and 0.028) compared to R0 and R1 resections, respectively. (Appendix A). Moreover, DTC detection was associated with worse PFS and OS (*p*-value = 0.004 and <0.001) compared to patients with no detectable DTCs (Appendix A). When taking only patients with R0 and R1 into consideration, HGC III–IV was associated with worse PFS (*p*-value < 0.001) and OS (*p*-value = 0.004) compared to patients with HGC I–II.

### 3.3. Multivariate Analysis

Using Cox Regression analysis, seven factors were correlated with overall survival (Table 3) and progression-free survival (Table 4) in the multivariate analysis. We analyzed the independent prognostic impact of the HGC on overall and progression-free survival by multivariate stratified analysis including age, gender, UICC TNM classification, tumor size, lymph node involvement, and resection margins. The HGC prognostic groups were identified as strong independent prognostic groups for overall survival (*p* = 0.001; HR 2.49, 95% CI, 1.47–4.22) and progression-free survival (*p* < 0.001; HR 2.87, 95% CI, 1.61–5.10). (Appendix A).

Data regarding postoperative Chemotherapy was available in 51 patients. Multivariate analysis of this subgroup of patients showed that HGC (*p*-value = 0.002) and UICC (*p*-value = 0.001) were independent prognostic factors for overall survival. Regarding progression-free survival, only HGC reached significance (*p*-value = 0.004). A detailed analysis is available in the Supporting data (Appendix A). However, these results should be regarded carefully due to the missing data and possible information bias.

## 4. Discussion

The results of this study show that the preoperative combination of disseminated tumor load in bone marrow indicated by DTC and systemic inflammation evaluated by GPS are associated with poor overall survival and progression-free survival in patients with PDAC. Furthermore, the preoperative HGC is a strong predictor of the oncological outcome of patients with PDAC and showed comparable survival stratification to the post-operative UICC TNM classification.

The TNM classification provided by the American Joint Committee of Cancer (AJCC) is the most used classification for PDAC. It is a well-defined cancer staging system that depends on the anatomic extension of the tumor [6,10]. However, unlike the HGC, accurate classification is only possible after histological examination of the specimen. As the majority of tumors (80–85%) do not undergo surgical resection, an exact pTNM classification is usually lacking [3].

Furthermore, TNM classification does not take into consideration other factors like tumor biology, or patient-related factors, which could explain why patients assigned the same stage may vastly vary in terms of tumor behavior and associated outcomes. This is supported by our findings, in that HGC stages are not associated with the degree of nodal involvement or with distant metastasis, but with progression-free survival and overall survival, which emphasizes the hypothesis that not only anatomic extension but also other biological factors can affect survival outcomes or that systemic micrometastasis is present early in the course of PDAC [16].

It is well known that even after R0 resection of PDAC and aggressive adjunct multimodal therapy, metastasis and tumor recurrence still occur [17]. A recent study compared patterns of recurrence between node-negative and node-positive resected PDAC and showed that distant metastasis is the most common site of tumor recurrence irrespective of the nodal status, which suggests that undetectable systemic micrometastases are present at the time of resection [16]. In the last decades, biomarkers for micrometastasis like circulating and disseminating tumor cells (CTC and DTC) were identified. Tumor dissemination in peripheral blood and bone marrow proved to be associated with shorter OS and PFS in different solid tumors and PDAC [4,5,18]. DTCs stay dormant in the bone marrow and recirculate into various sites causing metastasis or even tumor recurrence [19]. DTCs can be preoperatively easily accessed through a bone marrow aspiration compared to operative lymph node sampling and biopsy of distant sites. In our cohort, no complications related to the bone marrow aspiration were detected.

The GPS is a marker for the patient’s systemic immune response and is based on two acute phase reactant proteins, C-reactive protein (CRP) and Albumin. An elevated GPS reflects a compromised cell-mediated immunity [20]. This inflammation is induced by proinflammatory cytokines released by the tumor [21]. CRP is a positive acute-phase reactant in inflammation. Elevated preoperative CRP level is an independent predictor for poor outcome in PDAC [22].

Hypoalbuminemia is also a well-known negative prognostic factor in cancer patients undergoing surgery. Albumin is a nutritional marker that identifies poor nutritional status in cancer patients and is a negative acute-phase reactant that decreases in inflammatory states indicating a compromised immune system [23,24]. Consequently, elevated GPS has proven to be associated with poor prognosis in several solid tumors [25,26]. In PDAC, GPS was related to shorter overall survival [8]. Therefore, GPS and DTC status were combined into the Hamburg-Glasgow Classification. HGC has been previously implemented in non-metastatic esophageal cancer, which resulted to be an independent preoperative prognostic indicator for overall and progression-free survival [13].

In 2017, the definition of borderline resectable pancreatic ductal adenocarcinoma (BR-PDAC) was expanded from solely depending on anatomic criteria to include biological factors like questionable metastatic disease and conditional host factors such as suboptimal performance status, or severe medical comorbidities [27]. For this reason, the HGC could serve as an objective tool to identify patients with radiologically resectable PDAC with favorable (HGC I/II) or worse clinical outcomes (HGC III/IV). So that patients with HGC I–II could proceed with surgical exploration and resection, however, in patients with HGC III–IV, a neoadjuvant approach could be offered to treat micrometastases and to test tumor biology before resection as well as to increase possible R0 resections which is an important prognostic factor in PDAC [28].

In this study, we showed that the HGC is a strong predictor of oncological outcome in PDAC and is associated with shorter OS and PFS in patients with PDAC. Hamburg-Glasgow classification seems to be an objective, easily available, and significant prognosticator for the survival of patients with PDAC which should be used in addition to the TNM classification. Even in non-metastatic PDAC, the HGC showed significant survival stratification between the groups HGC I/II and III/IV, so that patients with resectable PDAC and HGC III/IV should be considered as borderline resectable PDAC, who could benefit from a neoadjuvant approach. Patients with HGC III/IV had a mean overall survival of 9.3 and 4.8 months, respectively, regardless of their TNM stage.

In this cohort, we included patients with different tumor stages, with localized and metastatic disease as well as patients in resectable and non-resectable settings, which simulates real-life situations that regularly encounter surgeons treating PDAC, where a preoperative prognostic classification is lacking and the decision for surgical exploration mostly depends on imaging studies. We were able to show that the HGC is independent of the TNM system. The implementation of the HGC into standard clinical preoperative staging might add significant information on tumor biology to the TNM classification, which might improve treatment in patients with PDAC. However, larger prospective validation studies are required before the implementation of HGC in TNM classification. Furthermore, our study could serve as a ground platform for the integration of new forms of liquid biopsies, like CTC, cfDNA, microRNA, and ctDNA [29,30] into clinical practice as a step-forward for personalized therapy in PDAC.

## 5. Limitations

The HGC is a promising staging classification system for PDAC. However, we recognize that there are limiting factors to our study: (i) single-institution study (ii) this study is from an era in which modern chemotherapy (such as gem nab-paclitaxel and FOLFIRINOX) for either adjuvant or systemic disease was not in use (iii) the lack of comparative data, for example, ASA or ECOG scores. However, we report on a homogenous and large study population in different stages of PDAC who had no preoperative systemic therapy which might affect the DTC and SIR status.

## 6. Conclusions

The HGC is a promising preoperative staging classification that shows significant risk stratification for overall survival and progression-free survival in patients with PDAC. Our results suggest that HGC can enable accurate preoperative staging in addition to the TNM classification which might improve the treatment of patients with PDAC.

## Figures and Tables

**Figure 1 cancers-13-05942-f001:**
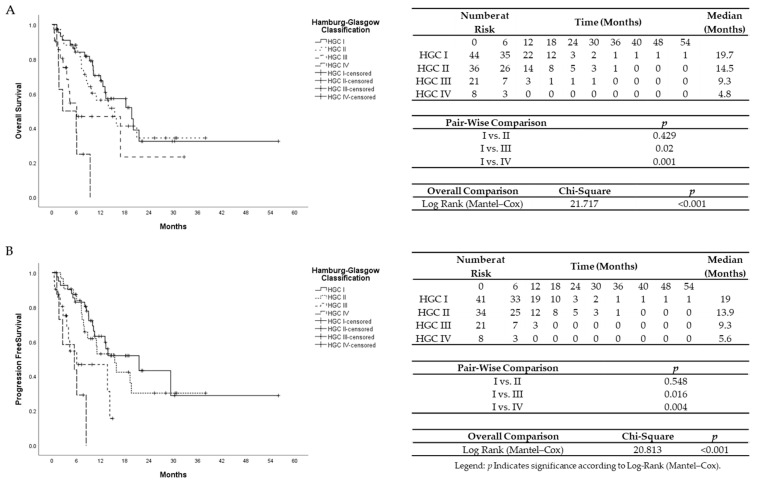
(**A**)Univariate Kaplan–Meier analysis for overall survival according to Hamburg-Glasgow classification. (**B**) Univariate Kaplan–Meier analysis for progression-free survival according to Hamburg-Glasgow classification.

**Figure 2 cancers-13-05942-f002:**
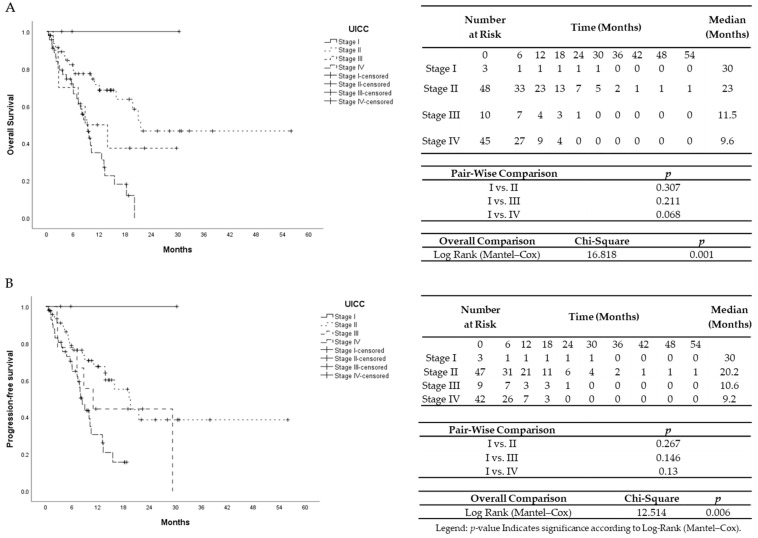
(**A**) Univariate Kaplan–Meier analysis for overall survival according to TNM UICC classification. **(B)** Univariate Kaplan–Meier analysis for progression-free survival according to TNM UICC classification.

**Figure 3 cancers-13-05942-f003:**
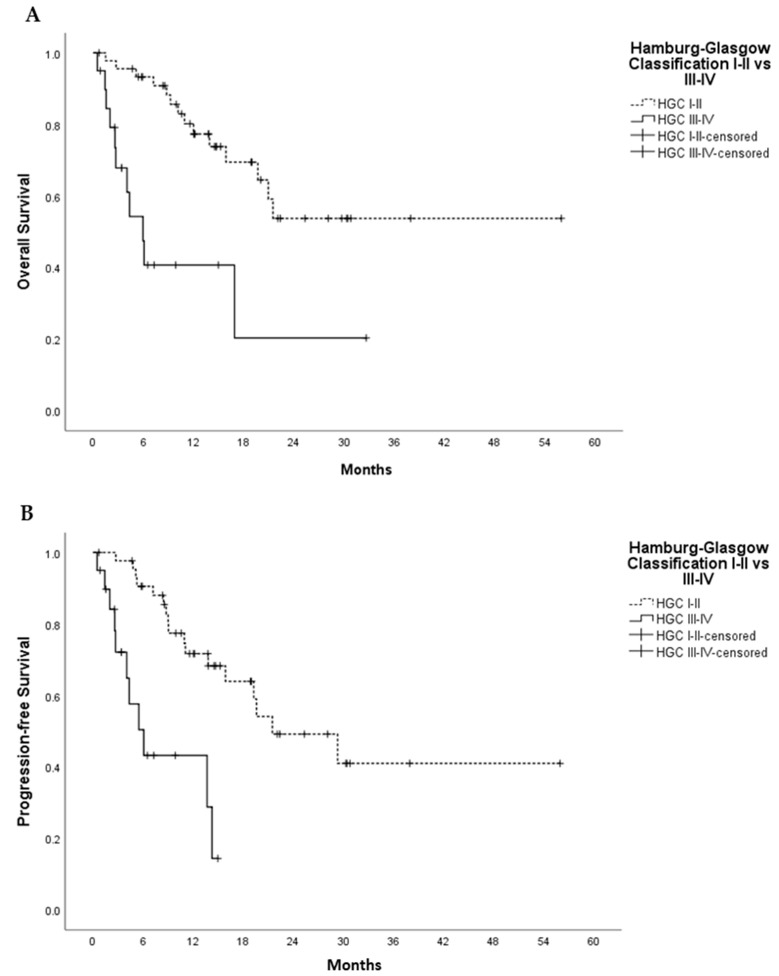
Univariate Kaplan–Meier analysis for overall survival (**A**) and progression free survival (**B**) in non-metastatic PDAC, HGC I–II vs. III–IV. In (**A**), Log Rank (Mantel-Cox) Chi-Square 14.456 and *p*-value < 0.001; In (**B**), Log Rank (Mantel-Cox) Chi-Square 16.405 and *p*-value <0.001. *p*-value indicates significance according to Log-Rank (Mantel–Cox) test.

**Table 1 cancers-13-05942-t001:** Definition of Hamburg-Glasgow Classification [13].

Variables	HGC I	HGC II	HGC III			HGC IV
DTC	Neg.	Neg.	Neg.	Pos	Pos	Pos
CRP	<10	≥10	≥10	<10	≥10	≥10
Albumin	and >35	or ≤35	and ≤35	and >35	or ≤35	and ≤35

Legend: CRP; C-reactive protein, DTC; disseminated tumor cell.

**Table 2 cancers-13-05942-t002:** Patient characteristics and the correlation between Hamburg-Glasgow classification (HGC) and clinicopathological parameters.

Variables	Hamburg-Glasgow Classification				Total	*p*
	HGC I	HGC II	HGC III	HGC IV		
Total	44	36	21	8	109	

Age						0.735
<65	20	17	7	4	48	
≥65	24	19	14	4	61	
					109	
Sex						0.638
Female	21	18	10	2	51	
Male	23	18	11	6	58	
					109	
Tumor grade						0.309
G1	3	2	0	0	5	
G2	21	16	11	1	49	
G3	11	7	5	4	27	
					81	
Tumor size						0.665
T1	0	1	0	0	1	
T2	3	0	1	0	4	
T3	24	22	14	3	63	
T4	11	8	4	1	25	
Tx	6	5	2	3	16	
					109	
Nodal status						0.837
N0	10	10	5	0	25	
N1	25	20	12	5	62	
Nx	7	6	3	2	18	
					105	
Metastatic status						0.415
M0	25	21	16	4	66	
M1	19	15	5	4	43	
					109	
Resection Margin						0.098
R0	16	12	6	3	37	
R1	10	11	9	0	30	
R2	4	4	1	4	13	
Rx (no resection)	12	8	4	2	27	
					109	
UICC						0.669
Stage I	2	0	1	0	3	
Stage II	15	17	12	3	47	
Stage III	5	3	1	1	10	
Stage IV	19	15	5	4	43	
					103	
Type of Operation						0.398
Distal pancreatectomy	6	5	1	1	13	
Pancreaticoduodenectomy	27	21	16	1	65	
Total pancreatectomy	2	1	0	1	4	
No resection	10	10	3	4	27	
					109	

Legend: *p*-value Indicates significance according to the Pearson Chi-square test between different HGC groups.

**Table 3 cancers-13-05942-t003:** Univariate and Multivariate analysis of overall survival in patients with PDAC.

	Univariate Analysis			Multivariate Analysis		
	HR	95% CI	*p*	HR	95% Cl	*p*
Age, <65, ≥65	1.55	0.89–2.67	0.115	1.88	1.04–3.41	0.035
Sex, male vs. female	1.01	0.59–1.74	0.994	1.37	0.76–2.45	0.288
Tumor size, T1–4	1.91	1.39–2.63	≤0.001	1.18	0.76–1.83	0.330
Nodal status, neg vs. pos	2.68	1.21–15.96	0.015	3.05	1.21–7.64	0.017
UICC, I–IV	1.51	1.22–1.86	≤0.001	1.30	0.98–1.73	0.064
HGC,			≤0.001			≤0.001
IV vs. I	0.16	0.06–0.39	≤0.001	0.13	0.04–0.46	≤0.001
IV vs. II	0.19	0.07–0.48	≤0.001	0.25	0.09–0.72	0.008
IV vs. III	0.37	0.14–1.01	0.053	0.78	0.24–2.4	0.67
Resection margin, R0/R1 vs. R2/Rx	2.51	1.45–4.37	0.001	1.51	0.79–2.90	0.21

Legend: *p* Indicates significance according to Cox regression analysis comparing the specified variables. HR indicates hazard ratio.

**Table 4 cancers-13-05942-t004:** Univariate and Multivariate analysis of progression-free survival in patients with PDAC.

	Univariate Analysis			Multivariate Analysis		
	HR	95% CI	*p*	HR	95% Cl	*p*
Age, <65, ≥65	1.66	0.95–2.90	0.074	2.08	1.13–3.85	0.019
Sex, male vs. female	0.993	0.57–1.71	0.978	1.29	0.71–2.31	0.392
Tumor size, T1–4	1.86	1.35–2.55	≤0.001	1.31	0.81–2.10	0.260
Nodal status, neg vs. pos	2.00	0.97–4.13	0.059	2.27	0.98–5.23	0.055
UICC, I–IV	1.51	1.22–1.86	≤0.001	1.13	0.84–1.51	0.412
HGC,			≤0.001			≤0.001
IV vs. I	0.16	0.06–0.44	≤0.001	0.13	0.04–0.41	≤0.001
IV vs. II	0.20	0.07–0.53	≤0.001	0.25	0.08–0.76	0.008
IV vs. III	0.51	0.18–1.41	0.197	0.93	0.28–3.04	0.90
Resection margin, R0/R1 vs. R2/Rx	2.92	1.65–5.17	≤0.001	2.03	1.04–3.98	0.038

Legend: *p* Indicates significance according to Cox regression analysis comparing the specified variables. HR indicates hazard ratio.

## Data Availability

The datasets used during the current study are available from the corresponding author on reasonable request.

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
