# Peer review of "Evaluation of the Hamburg-Glasgow Classification in Pancreatic Cancer: Preoperative Staging by Combining Disseminated Tumor Load and Systemic Inflammation"

_cancers, 2021, doi:10.3390/cancers13235942_

Round 1
Reviewer 1 Report
The study of Abdalla et al. compares the novel “Hamburg Glasgow Classification” (HGC) with the UICC classification in patients with resectable pancreatic ductal adenocarcinoma (PDAC) in order to evaluate its potential as prognostic tool for a better patient stratification.
Overall, the study demonstrates a novel classification system integrating CRP and albumin levels from the Glasgow Prognostic Score and assessment of disseminated tumor cells (DTC) in bone marrow of PDAC patients prior to surgical removal of the primary tumor. The HGC score was correlated with various clinico-pathological parameters in an univariate analysis as well as in a multivariate analysis.
The study is well written and addresses an interesting and novel issue in order to improve the treatment of PDAC patients.
However, some points have to be raised that need to be clarified:
- In Chapter 2.1 information regarding the number of patients with and without adjuvant chemotherapy as well as which kind of adjuvant chemotherapy was applied should be given.
- The resection margin does not correlate with the HGC. However, there is apparently a high variance regarding the HGC within each group. Thus, do R0 patients with a lower HGC survive longer than R0 patients with a higher HGC?
- The discussion is in large parts rather comparable to an introduction. The discussion would clearly benefit from a more critical reflection of the findings, e.g. it can be speculated why HGC does not correlate with nodal status or metastasis?
- The authors suggest in the discussion that HGC “seems to be an […] significant prognosticator for the survival of patients with PDAC which should be used in addition to the TNM classification.” The author should outline in more detail the (clinical) benefit of combining both classification systems.
- In Figure 3, the labelling of subfigures A+B is missing. Furthermore, in the table below “overall survival” should be added to Figure A and “progression free survival” should be added to Figure B.
- I recommend a consistent use of PDAC throughout the paper and not a mix between pancreatic cancer and PDAC.
Author Response
Response to Reviewer 1 comments:
First of all we would like to give our thanks to the Reviewer 1 on his thoughtful comments and suggestions.
- In Chapter 2.1 information regarding the number of patients with and without adjuvant chemotherapy as well as which kind of adjuvant chemotherapy was applied should be given.
Thank you for this remark. Data regarding postoperative chemotherapy was only available in 51 patients. Due to the small number of patients, subgroup analysis for this set of patients has been added separately in the supplementary information. Details regarding the chemotherapy regimen, dosage, number of cycles, and response to chemotherapy are not available. This information has been added in Chapter 2.1, as requested (line 92-95).
- The resection margin does not correlate with the HGC. However, there is apparently a high variance regarding the HGC within each group. Thus, do R0 patients with a lower HGC survive longer than R0 patients with a higher HGC?
Thank you, we addressed this remark and conducted a univariate analysis using Kaplan-Meier analysis, when considering only R0 (n=37 ), the univariate analysis shows that there HGC I/II tends to have a longer OS (p=0.083) but no difference regarding PFS (0=0.268). To make a more robust analysis we took both R0 and R1 (n=67) toghether and compared HGC I-II vs III-IV, here was HGC I-II associated with longer PFS and OS (<0.001 and =0.004 respectively) compared to HGC III-IV. This was added in Chapter 3.2 (line 165-167).
- The discussion is in large parts rather comparable to an introduction. The discussion would clearly benefit from a more critical reflection of the findings, e.g. it can be speculated why HGC does not correlate with nodal status or metastasis?
In our findings, HGC is not associated with nodal status or distant metastasis. However, the survival outcomes clearly are associated with both with M and HGC. This shows, that not only tumor extension defined by positive nodal status or distant metastases is prognostic for survival but also, that tumor biology and patient performance are important prognostic factors. This has been added in the discussion (line 219-226).
- The authors suggest in the discussion that HGC “seems to be an […] significant prognosticator for the survival of patients with PDAC which should be used in addition to the TNM classification.” The author should outline in more detail the (clinical) benefit of combining both classification systems.
Combining both TNM and HGC might identify patients who could benefit from primary resection or patients who would rather benefit from neoadjuvant therapy. In cases, where imaging studies suggest a primary resectable pancreatic cancer without distant metastases but with unfavorable HGC (III-IV), an approach similar to borderline-resectable pancreatic cancer could be chosen, starting with neoadjuvant therapy and considering surgery depending on tumor response. This has been added in the discussion (line 255-263)
- In Figure 3, the labelling of subfigures A+B is missing. Furthermore, in the table below “overall survival” should be added to Figure A and “progression free survival” should be added to Figure B.
The labeling was added to figure 3 it’s table accordingly.
- I recommend a consistent use of PDAC throughout the paper and not a mix between pancreatic cancer and PDAC
As you requested, the term pancreatic cancer has been changed throughout the manuscript. It remained unchanged in the title and supplementary material.

Reviewer 2 Report
The manuscript entitled "Evaluation of the Hamburg-Glasgow Classification in pancreatic cancer: Preoperative staging by combining disseminated tumor load and systemic inflammation" deals with an extremely interesting topic, pancreatic cancer is difficult to detect and stage and often when it is now diagnosed for patients it is too late.
The manuscript is well structured, even the references adequately support the subject matter.
However, I have some considerations:
patients were dosed with c-reactive protein, consequently the authors have both blood and serum of these patients.
Consequently why not evaluate cfDNA ??
In recent years, the evaluation of cfDNA in the detection and staging of pancreatic cancer has had great resonance, as the changes in free circulating DNA would have preceded the appearance of the tumor in its most aggressive form.
Therefore, where it is not possible to carry out this evaluation (although I suggest it) at least to discuss the application of this dosage in the discussion as a future perspective, I recommend the following works (doi: 10.3390 / genes11010014; doi: 10.1186 / s13148-019-0728 -8, DOI: 10.1038 / bjc.2017.219)
it would be interesting to intertwine the data obtained by the authors with the cfDNA assay.
Author Response
Response to Reviewer 2 comments:
- patients were dosed with c-reactive protein, consequently the authors have both blood and serum of these patients.Consequently why not evaluate cfDNA ??
We would like to give thanks to Reviewer 2 for his comment. By the time this study was conducted, we performed at our institution many studies regarding the detection DTC in the bone marrow for many tumor entities, among them pancreatic cancer but not cfDNA or ctDNA.
We agree that the integration of other types of liquid biopsy like CTC, cfDNA and ctDNA with clinical parameters could deliver interesting results in the future.
As requested, this has been added in the discussion (line 283-285).